# COVID-19 Vaccine Acceptance and Associated Factors among Women in Saudi Arabia: A Cross-Sectional Study

**DOI:** 10.3390/vaccines10111842

**Published:** 2022-10-31

**Authors:** Noor Alshareef

**Affiliations:** 1Department of Health Services and Hospital Administration, Faculty of Economics and Administration, King Abdulaziz University, Jeddah 22254, Saudi Arabia; noor_alshareef@hotmail.com; 2Health Economics Research Group, King Abdulaziz University, Jeddah 22254, Saudi Arabia

**Keywords:** COVID-19, vaccine, vaccine acceptance, women, Saudi Arabia, female population, hesitancy

## Abstract

Although women have been substantially affected by the pandemic, they tend to have a lower likelihood of COVID-19 vaccine acceptance. Research on factors associated with COVID-19 vaccine acceptance among this key population is imperative. Thus, this study aimed to assess COVID-19 vaccine acceptance and associated factors among women in Saudi Arabia. This study was part of a larger study conducted on the acceptance of the COVID-19 vaccine in Saudi Arabia, carried out between the 8th and 14th of December 2020. The study sample included 910 women aged 18 years and older. Bivariate and multivariable regression analyses was utilized to analyze the data. Overall, 41% of the participants were willing to receive the vaccine. Participants were more willing to accept vaccination if they were 40–49 years old (OR = 2.209, 95% CI: 1.49–2.02), if they had a moderate (OR = 2.570, 95% CI 1.562–4.228) or high to very high (OR = 1.925, 95% CI 1.093–3.390) perceived likelihood of being infected with COVID-19, or if they were in favor of mandatory COVID-19 vaccination for people in Saudi Arabia (OR = 64.916, 95% CI 35.911–117.351). However, participants with a high educational level (OR = 0.431, 95% CI 0.220–0.847) or who refused physician-recommended vaccines in the past (OR = 0.152, 95% CI 0.083–0.275) were less likely to accept COVID-19 vaccination. Given the low level of vaccine acceptance among women, relevant stakeholders should consider the needs and dynamics of this key population to increase vaccination uptake and to improve current and future outreach activities.

## 1. Introduction

Vaccine development is an effective weapon against the COVID-19 pandemic. For centuries, vaccines have been used to fight against infectious diseases, including measles and influenza [1,2]. The use of vaccination as a prevention strategy can substantially lower the strain on healthcare resources caused by fast-spreading epidemics and pandemics. Toward the end of December 2020, the World Health Organization (WHO) began emergency-use authorization for SARS-CoV-2 vaccines [3]. With the approval of the COVID-19 vaccine, a further layer of protection that was dependent on uptake rather than human behavior (e.g., wearing a face mask) became available [4]. 

Uptake is a critical component for the success of any vaccination program. To realize the benefits of herd immunity at the population level, a threshold proportion of the population must receive vaccination [5,6]. This threshold has been estimated to be about 67% for COVID-19 vaccines [6,7]. However, although authorized and available vaccines are effective in protecting individuals from serious illness, the acceptance of vaccination varies worldwide from 56.9% in the USA and 48% in the Kingdom of Saudi Arabia (KSA) to 23.6% in Kuwait [8,9]. Low acceptance rates among the general population or among certain segments of the population threaten the success of vaccination programs. Globally, COVID-19 vaccination campaigns have faced a high level of vaccination hesitancy [7,8], which is defined as “a state of indecision and uncertainty that precedes a decision to become (or not become) vaccinated” [10]. Vaccination hesitancy is a complex process with different interplaying factors that are dependent on socioeconomics and cultural context [4]. Rises in vaccine hesitancy often coincide with newly reported vaccine risks, new information, or new policies [10].

COVID-19 does not equally influence women and men, with men being at higher risk [11]. Mortality rates among men who contracted the virus were 40% higher than those among women, and intensive care admission was also three times greater among men [12]. Women’s health, however, is adversely impacted by declining access to sexual and reproductive health services [13]. Research from infectious-disease-driven economic crises indicates a substantial impact on women. For instance, the 2014–2016 West African Ebola outbreak indicated that women suffered more throughout the pandemic, first because of their roles as caregivers, which resulted in higher infection rates among women [14,15], and second, because the types of jobs that were more often held by women (retail trade, tourism, and hospitality) were seriously affected by the economic downswing [16]. 

During the COVID-19 outbreak, the physical load and pressure experienced by women were often greater than those by men [1]. Women played an important role in healthcare responses to the COVID-19 crisis [17]. At the same time, they faced a compounding burden; they perform most of the unpaid care and domestic work and face a high risk of economic insecurity, exploitation, violence, abuse, and harassment during quarantines or crises [17]. In the KSA, Qattan (2022) found that, of the 1527 women that were surveyed, 36% reported symptoms of mild psychological distress, and 8% experienced severe distress levels during the outbreak [18]. Similarly, Liu et al. showed that women reported higher post-traumatic stress symptoms after one month of the COVID-19 outbreak in Wuhan, China. They also found that women suffered more re-experiencing, negative alterations in cognition or mood, and hyperarousal sub-symptoms than males [19]. 

Given these challenges, coupled with the existing literature on high vaccine hesitancy among women [9,20,21], this study attempted to examine vaccine acceptance among women. Although other researchers have attempted to understand vaccine acceptance among women, specifically focusing on pregnant women and women of reproductive age, little is known about COVID-19 vaccine hesitancy among women in the KSA. Therefore, this study, aimed to examine the factors associated with the acceptance of a future COVID-19 vaccine among women and the underlying reasons for vaccine hesitancy among this population. While most government policies affect both sexes, special attention must be devoted to the women to create tailored sex-specific strategies that address specific concerns while considering specific contexts to enhance the national vaccination program’s success. This study was part of a larger study that was conducted to study the willingness among the public in the KSA to accept COVID-19 vaccination [9,21,22]. At the time of the survey, only the Pfizer–BioNTech COVID-19 vaccine had been approved in the country however, the national vaccination campaign had not yet been launched.

## 2. Materials and Methods

This cross-sectional online study was carried out between the 8 and 14 of December 2020 (at the time, the Pfizer vaccine was approved in Saudi Arabia). The eligibility criteria required that participants were currently living in Saudi Arabia and were 18 years old or older. 

Participants completed the survey online using Survey Monkey. Using a simplified snowball sampling technique, participants were recruited online using social media platforms and were requested to share the survey link with their contacts. Those who agreed to participate in the study filled in the questionnaire on their electronic devices. Before answering the questionnaire, the participants received an electronic informed consent form. This form informed the participants about the purpose of the study and its voluntary and anonymous nature. A total of 2319 participants enrolled in this study, and 2137 completed the survey [9]. The overall completion rate of the questionnaire was 92.15%. However, this study was only limited to women participants (910). A more detailed description of the sampling method can be found elsewhere [9,21].

The self-administered questionnaire was adopted from earlier research studies on vaccination behaviors and was validated by a panel of experts [23,24,25,26,27]. The instrument was translated from English to Arabic and then back-translated; however, the survey was conducted in Arabic. The first section included questions about participants’ sociodemographic characteristics. The second section included questions about participants’ health status, immunization history, COVID-19 infections among family and friends, the perceived risk of COVID-19, and the possibility of contracting COVID-19. The third section included questions about participants’ acceptance of COVID-19 vaccination, including reasons for not accepting the COVID-19 vaccine. 

### Variables

In this study, the primary focus was COVID-19 vaccine acceptance among women. Vaccine acceptance was assessed by using the following question: “Scientists around the world are currently working on a vaccine that could prevent people from getting infected with COVID-19. It is hoped that the vaccine will become available in a few months. In the case that a COVID-19 vaccine becomes available in the next few months, with an effectiveness rate of between 90 and 94.5%, would you be willing to get the COVID-19 vaccine if it was provided for free by the government?” The response options were “yes” and “no”. For participants who were not willing to receive the vaccine, the questionnaire asked about their reasons for refusal in a multiple-choice question with 8 options: “ Fear of adverse side effects”, “Safety and efficacy concerns”, “The speed of making the vaccine”, “The short duration of clinical trials”, “Personal preference to not get vaccinated”, “Belief that the vaccine is a plot”, “Belief that the virus does not exist”, “Belief that masks and sanitizers are sufficient for protection”, and “Other”. 

The study also examined the factors that contribute to vaccination acceptance, including demographic characteristics such as age, marital status, educational level, and employment status. Age was divided into five categories: 18–29 (the reference category), 30–39, 40–49, 50–59, and ≥60 years old. Marital status was recorded as a binary value; a value of one for married was used and zero otherwise (single, widowed, or divorced). Educational level was categorized into three groups: high school or below (reference group), bachelor’s degree, and postgraduate degree. Employment status was also divided into six groups: government employee (reference group), private sector employee, self-employed, student, retired, and unemployed. 

The questionnaire also asked participants about whether they (1) had a chronic disease that would place them at a higher clinical risk of severe illness from COVID-19; (2) had been vaccinated for seasonal influenza in the past; (3) had ever refused a vaccine recommended by a physician because of doubts concerning the vaccine; (4) had a history of COVID-19 infection among their family members and friends; and (5) had lost a family member or friend due to complications from COVID-19. The response options for these five questions were “yes” and “no”. In addition, the questionnaire asked about the participants’ perceptions regarding the extent to which they think COVID-19 poses a risk to people in the country, and the response options were “minor risk or no risk”, “moderate risk”, and “significant or major risk”. Moreover, the questionnaire asked about their perceptions of the likelihood of becoming infected with COVID-19, and the response options were “low or very low”, “fair”, and “high or very high”. In addition, the questionnaire asked about the participants’ support for mandatory COVID-19 vaccination, and the responses were binary outcomes of “yes” and “no”.

Bivariate analysis was used for cross-tabulation between all variables and the independent variable of interest using chi-squared tests. Multivariable logistic regression analysis was conducted to identify and examine the variables correlated with COVID-19 vaccine acceptance, with calculations of the odds ratio (OR) and a 95% confidence interval (CI). I processed this study’s data and conducted the analysis using the STAT 15.1 software (StataCorp LP, College Station, TX, USA). 

## 3. Results

The socio-demographic characteristics of this study’s participants are presented in Table 1. Of the surveyed women, 29.67% were 18–29 years old, 32.09% were 30–39 years old, 17.91% were 40–49 years old, 12.86% were 50–59 years old, and 7.47% were 60 years old or above. Nearly 61% of the woman participants were married, and half of them had a university degree. Regarding employment status, 33.41% of the women worked in the public sector, and 30.88% were unemployed. Regarding health status, 22.64% of participants reported having chronic conditions. Moreover, in the past, 73.08% of participants had not refused any vaccination, and half of them had received the flu vaccine. More than half of the participants had no family members who had been infected with COVID-19, and 86% had a friend who had been infected with COVID-19. Moreover, only 26% of participants had lost a family member or friend due to COVID-19 complications.

Regarding the perceived risk of COVID-19, a large portion (43.85%) of the women participants perceived a significant or major risk, and 35.16% perceived a moderate risk, as shown in Table 1. However, a large portion of participants (41.21%) had low or very low levels of concern regarding being infected with the virus, and 27.25% had high or very high levels of concern. In addition, the majority of women participants (70%) believed that the COVID-19 vaccine should not be compulsory for citizens and residents.

As shown in Table 1, the following were all statistically significant: age, marital status, educational level, employment status, having received the flu vaccine in the past, having refused vaccination in the past, the perceived risk of COVID-19, concerns about being infected with COVID-19, and the belief that the COVID-19 vaccine should be compulsory for all citizens and residents of Saudi Arabia. 

Table 2 presents the results of the logistic regression analysis of factors associated with COVID-19 vaccine acceptance among women. Participants who were 40–49 years old were more likely to accept vaccination against COVID-19 compared with participants who were 18–29 years old (OR: 2.209; 95% CI: 1.019–4.789). Women with a postgraduate educational level were less likely to accept the vaccine compared with women with a secondary school or lower education (OR: 0.431, 95% CI: 0.220–0.847). Moreover, compared with women working in the public sector, women students were more willing to accept the vaccine (OR: 2.285; 95% CI: 0.990–5.274). Women participants who refused a vaccine recommended by a physician in the past were less likely to accept the COVID-19 vaccine (OR: 0.152; 95% CI: 0.083–0.275). In addition, compared with women participants with low or very low levels of concern about becoming infected with COVID-19, those with high or very high levels of concern were more likely to accept the COVID-19 vaccine (OR: 1.925; 95% CI: 1.093–3.390). Women who believed that the COVID-19 vaccine should be compulsory for all citizens and residents were more likely to accept the vaccine compared with those who did not support mandatory vaccination (OR: 64.916; 95% CI: 35.911–117.351).

Table 3 presents reasons for the lack of willingness among women to be vaccinated. “Fear of adverse side effects” emerged as the main reason for the lack of willingness to vaccinate, as expressed by 29.74% of participants. Only 1.30% of participants believed that the virus did not exist. Other cited reasons included “Safety and efficacy concerns” (21%), “Short duration of clinical trials” (13.57%), “Personal preference to not get vaccinated” (8.55%), and “Belief that the vaccine is a plot” (7.43%).

## 4. Discussion

Understanding women’s attitudes toward accepting a COVID-19 vaccine and the factors that are likely to predict their willingness to do so, highlights that different segments of the population have different sentiments and perspectives toward COVID-19 vaccines and, consequently, may have different determinants of acceptance. Among the women population, different factors that drive vaccine acceptance have arisen. Being 40–49 years of age, having a moderate or high perceived likelihood of being infected with COVID-19, and being in favor of mandatory vaccination were associated with COVID-19 vaccine acceptance. However, women with a high educational level and those who had refused physician-recommended vaccines in the past were less likely to accept COVID-19 vaccination. A better understanding of the predictors of vaccine acceptance among this specific portion of the population, which is more likely to be unvaccinated, can assist health authorities and relevant stakeholders in considering the needs and dynamics of women when crafting strategies to enhance vaccine uptake, as informed by contextualized research. 

This study examined women willingness to receive the COVID-19 in the KSA. The results indicated that, although 41% of women were willing to receive the vaccine, more than half of the women participants (59%) were not. The vaccine acceptance rate among women in this study was generally similar to the rates found in the general population, which was 48% during the same period in Saudi Arabia [9]. In this study, the women participants who were not willing to receive vaccination reported concerns surrounding safety and efficacy as well as their fear of adverse side effects, which were the main reasons for their lack of willingness to receive the vaccination. This showed that safety-related issues are a key consideration in an individual’s decision making. During outbreak crises, questions concerning new vaccines, newly reported research about the vaccine, and vaccination campaigns arise [10]. The speed of vaccine development, its technological novelty, and the effectiveness of the vaccine all amplify distrust [28]. Therefore, designing and planning public health communications that ensure vaccination uptake and taking measures to garner trust in the safety and efficacy of the vaccine are paramount.

Women are often the gatekeepers to their families’ health decisions [28]. Women also play an important role in influencing the pandemic. Given their mobility and active role in taking care of their families, they can substantially contribute to the spread of the virus. Additionally, the disproportionate economic impact that crises have on women result in instability and social challenges for their families. What they expose themselves to on social media and what they learn from their friends can immensely impact their decisions [4]. Women, especially those of reproductive age, are a particular subset of the population who, when making decisions surrounding vaccine uptake, consider themselves and their fertility. Several factors, including future fertility, pregnancy, and breastfeeding, are unique factors driving vaccination hesitancy in women of childbearing age [4]. This study found that women between the age of 40 and 49 years were more likely to accept the vaccine compared with women between 18 and 29 years old. While, both categories fall within the reproductive age bracket of 15–49 years old [4], women aged 40–49 were more willing to accept vaccination compared with women below or above this age. Although this study did not investigate whether these women were pregnant or breastfeeding their babies, this finding might indicate that women who were less willing to receive the vaccine were pregnant or breastfeeding. A recent study conducted in the KSA reported that pregnant women and those planning on being pregnant were more hesitant to be vaccinated than women who were not pregnant or not planning to be pregnant [29]. 

Support of mandatory vaccination was strongly correlated with vaccine acceptance among women. This finding could be a reflection of the pandemic’s unique circumstances or experiences that influenced them. The harsh reality of the COVID-19 pandemic and the morbidity and mortality associated with it have substantially impacted communities, which, to an extent, enhanced their readiness to accept the vaccine. Positive vaccine acceptance among women reflects their high levels of empathy toward the safety of families and communities and their desire to return to their previous work regimen and childcare arrangements [28]. Furthermore, this study found that the perceived likelihood of contracting COVID-19 could impact vaccine acceptance, as women with a moderate or high perceived likelihood of contracting the virus had an increased likelihood of accepting the COVID-19 vaccine. This is consistent with similar findings among healthcare workers and the general population [9,21,30] and indicates that a COVID-19 vaccine is seen as effective protection against the virus.

This study also examined vaccination history, assuming that prior vaccination decisions predict future decisions. Prior vaccinations may infer a trust in vaccines in general and an understanding of their value and benefits [31]. The study also found that past acceptance of physician-recommended vaccination had a lower COVID-19 vaccine acceptance rate, which contrasts the findings of a previously reported study conducted among the Italian population [30]. In this study, women’s attitudes toward vaccination practices tended to be consistent among all vaccines, including the COVID-19 vaccine. Vaccine acceptance is an individual habit that might apply to vaccines for different diseases that share similar clinical characteristics and modes of transmission [32,33]. 

Higher levels of vaccine and health literacy have been correlated with higher levels of education [28]; however, this study’s results showed that higher education was associated with a lower likelihood of receiving the COVID-19 vaccine. This finding suggested that high educational attainment on its own is not a determinant of vaccine acceptance and that, according to Biasio (2016), general education does not necessarily indicate health literacy [34]. Thus, more efforts to improve confidence and trust in the vaccine among women with higher educational achievement are critical. 

To the best of my knowledge, this is the first study to assess vaccine acceptance and its determinants among women in the KSA. However, this study has some limitations. The cross-sectional study design used in this study provides a snapshot of the period in which I conducted it, and causal relationships cannot be inferred. Moreover, the outcomes of the self-administrated questionnaire may have been affected by social desirability bias, misclassification bias, and misreporting. Moreover, because I used convenience sampling in this study, caution should be taken when applying this study’s results to other populations or settings. A further limitation is that the online nature of this study may have impacted the generalizability of the results and may have limited access to more participants. 

## 5. Conclusions

The findings of this study suggested that predictors of vaccine acceptance among women differ from those of the general population. This study represents an initial attempt to delineate the extent of the challenges related to vaccine acceptance among women in particular, and it underlines that a “one size fits all” approach may not be appropriate with respect to enhancing vaccine uptake and building trust in COVID-19 vaccination. As women are the gatekeepers of their families’ health, developing sex-specific communication strategies and mitigating vaccine hesitancy is important. 

## Figures and Tables

**Table 1 vaccines-10-01842-t001:** Frequency distribution and chi-squared analysis of acceptance of COVID-19 vaccination among women (*n* = 910).

Variable	Not Willing to Accept COVID-19 Vaccination *n* = 538 59.12%		Willing to Accept COVID-19 Vaccination *n* = 372 40.88%		Total (%) *n* = 910		*p*-Value
	A		B		C		D
Age (years)							
18 to 29	134	24.91	136	36.56	270	29.67	<0.001 ***
30 to 39	187	34.76	105	28.23	292	32.09	
40 to 49	98	18.22	65	17.47	163	17.91	
50 to 59	74	13.75	43	11.56	117	12.86	
≥60	45	8.36	23	6.18	68	7.47	
Marital status							
Unmarried	184	34.20	172	46.24	356	39.12	<0.001 ***
Married	354	65.80	200	53.76	554	60.88	
Educational level							
High school or below	101	18.77	120	32.26	221	24.29	<0.001 ***
Bachelor’s degree	271	50.37	185	49.73	456	50.11	
Postgraduate degree	166	30.86	67	18.01	233	25.60	
Employment status							
Government employee	194	36.06	110	29.57	304	33.41	<0.001 ***
Private sector employee	44	8.18	26	6.99	70	7.69	
Self-employed	20	3.72	9	2.42	29	3.19	
Student	53	9.85	94	25.27	147	16.15	
Retired	57	10.59	22	5.91	79	8.68	
Unemployed	170	31.60	111	29.84	281	30.88	
Suffer from chronic disease							
No	418	77.70	286	76.88	704	77.36	0.773
Yes	120	22.30	86	23.12	206	22.64	
Received flu vaccine in past							
No	306	56.88	156	41.94	462	50.77	<0.001 ***
Yes	232	43.12	216	58.06	448	49.23	
Refused vaccination in past							
No	324	60.22	341	91.67	665	73.08	<0.001 ***
Yes	214	39.78	31	8.33	245	26.92	
Family member(s) infected with COVID-19							
No	304	56.51	223	59.95	527	57.91	0.301
Yes	234	43.49	149	40.05	383	42.09	
Friend(s) infected with COVID-19							
No	78	14.50	45	12.10	123	13.52	0.298
Yes	460	85.50	327	87.90	787	86.48	
Lost family member or friend due to complications from COVID-19							
No	410	76.21	262	70.43	672	73.85	0.051 *
Yes	128	23.79	110	29.57	238	26.15	
Perceived risk of COVID-19							
Minor or no risk	144	26.77	47	12.63	191	20.99	<0.001 ***
Moderate risk	203	37.73	117	31.45	320	35.16	
Significant or major risk	191	35.50	208	55.91	399	43.85	
Concerned about becoming infected with COVID-19							
Low or very low	271	50.37	104	27.96	375	41.21	<0.001 ***
Fair	153	28.44	134	36.02	287	31.54	
High or very high	114	21.19	134	36.02	248	27.25	
COVID-19 vaccine should be compulsory for all citizens and residents							
No	521	96.84	115	30.91	636	69.89	<0.001 ***
Yes	17	3.16	257	69.09	274	30.11	

*** *p* < 0.01, * *p* < 0.1.

**Table 2 vaccines-10-01842-t002:** Logistic regression estimates of factors associated with acceptance of a COVID-19 vaccine among women (*n* = 910).

Variable	OR	95% CI	*p*-Value
Age (years)			
18 to 29 (ref)			
30 to 39	1.259	0.625–2.534	0.519
40 to 49	2.209	1.019–4.789	0.045 **
50 to 59	1.474	0.611–3.559	0.388
≥60	2.129	0.752–6.027	0.155
Marital status			
Unmarried (ref)			
Married	0.817	0.500–1.336	0.421
Educational level			
High school or below (ref)			
Bachelor’s degree	0.654	0.392–1.093	0.105
Postgraduate degree	0.431	0.220–0.847	0.015 **
Employment status			
Government employee (ref)			
Private sector employee	0.885	0.375–2.088	0.780
Self-employed	0.887	0.224–3.506	0.864
Student	2.285	0.990–5.274	0.053 *
Retired	0.548	0.206–1.453	0.226
Unemployed	1.056	0.582–1.916	0.858
Suffer from chronic disease			
No (ref)			
Yes	0.838	0.501–1.403	0.502
Received flu vaccine in past			
No (ref)			
Yes	1.016	0.671–1.538	0.942
Refused vaccination in past			
No (ref)			
Yes	0.152	0.083–0.275	<0.001 ***
Family member(s) infected with COVID-19			
No (ref)			
Yes	1.318	0.852–2.040	0.215
Friend(s) infected with COVID-19			
No (ref)			
Yes	1.110	0.478–1.665	0.719
Lost family member or friend due to complications from COVID-19			
No (ref)			
Yes	1.167	0.723–1.883	0.527
Perceived risk of COVID-19			
Minor or no risk (ref)			
Moderate risk	1.047	0.566–1.939	0.883
Significant or major risk	1.266	0.679–2.363	0.458
Concerned about becoming infected with COVID-19			
Low or very low (ref)			
Fair	2.570	1.562–4.228	<0.001 ***
High or very high	1.925	1.093–3.390	0.023 **
COVID-19 vaccine should be compulsory for all citizens and residents			
No (ref)			
Yes	64.916	35.911–117.351	<0.001 ***

*** *p* < 0.01, ** *p* < 0.05, * *p* < 0.1.

**Table 3 vaccines-10-01842-t003:** Reasons for not accepting the COVID-19 vaccination.

Reason	N	%
Fear of adverse side effects	160	29.74
Safety and efficacy concerns	113	21.00
The speed of making the vaccine	23	4.28
The short duration of clinical trials	73	13.57
Personal preference to not get vaccinated	46	8.55
Belief that the vaccine is a plot	40	7.43
Belief that the virus does not exist	7	1.30
Belief that masks and sanitizers are sufficient for protection	32	5.95
Other	44	8.18
Total	538	100

## Data Availability

The datasets generated and/or analyzed in this current study are not publicly available due to privacy and confidentiality agreements as well as other restrictions, but they are available from the corresponding author (NA) on reasonable request.

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
