# Peer review of "COVID-19 Vaccine Acceptance and Associated Factors among Women in Saudi Arabia: A Cross-Sectional Study"

_vaccines, 2022, doi:10.3390/vaccines10111842_

Round 1

Reviewer 1 Report

Tittle

COVID -19 Vaccine Acceptance and Associated Factors among 2

Women in Saudi Arabia

Comment: could add in the tittle what sort of study is this?  To read as

COVID -19 Vaccine Acceptance and Associated Factors among 2

Women in Saudi Arabia: A cross-sectional study

Keywords

Comment: you could hesitancy

Introduction

The introduction is well written, however should by narrate the study objective, which I see in the methodology.

Comment: This study was part of a larger study conducted to study the willingness to accept a 88

COVID-19 vaccine among the public in KSA. Could you shift this para to end of the introduction

Methods

A total of 2319 participants enrolled in this study, and 2137 completed the survey.

Comment: add response rate%

Comment: Provide the source of the tool survey using and whether the tool was validated and by whom?.

Comment: was the female include in the study Saudi national only or both national and non-national?

Results

Comment: whenever you inform the 5 , you should N=? for the figures. ie. There were 29.67% (N=?0 female aged 18-29, 32.09% (N=?) aged 30-39, 17.91% aged

Discussion

Comment: at the beginning of the discussion narrate the key study findings.

Author Response

Response to Reviewer 1 Comments

Point 1: Could add in the title what sort of study is this?  To read as COVID -19 Vaccine Acceptance and Associated Factors among Women in Saudi Arabia: A cross-sectional study

Response 1: Thank you for raising this comment.  The study title is modified to “COVID -19 Vaccine Acceptance and Associated Factors among Women in Saudi Arabia: A cross-sectional study”

Point 2: Comment: you could hesitancy

Response 1: Thank you for your comment.  The keyword “hesitancy” is added. Line 26

Point 3: The introduction is well written, however should by narrate the study objective, which I see in the methodology.

Response 1: Thank you for your comment. The objective of the study is mentioned in the introduction section. Lines 78-81

Point 4: This study was part of a larger study conducted to study the willingness to accept a 88

COVID-19 vaccine among the public in KSA. Could you shift this para to end of the introduction

Response 1: Thank you for your comment. As suggested this part was moved to the end of the introduction section

Point 5: A total of 2319 participants enrolled in this study, and 2137 completed the survey.

Comment: add response rate%

Response 1: Thank you for your comment. The completion rate was added. Lines 99-100.

Point 6: Provide the source of the tool survey using and whether the tool was validated and by whom?

Thank you for your comment.  the source of the tool is referenced and the tool is validated by a panel of experts in the field “The self-administrated questionnaire was adopted from earlier research on vaccination behaviors and was validated by a panel of experts. Lines 104-105

Point 7: was the female include in the study Saudi national only or both national and non-national?

Response 1: Thank you for your comment. Yes, all females who took part in the study regardless of nationality were included in the study.

Point 8: whenever you inform the 5, you should N=? for the figures. ie. There were 29.67% (N=?0 female aged 18-29, 32.09% (N=?) aged 30-39, 17.91% aged.

Response 1: Thank you for your comment. This was changed to “There were 29.67% females aged 18-29, 32.09% aged 30-39, 17.91% aged 40-49, 12.86% aged 50-59, and 7.47 aged 60 years old or above.”. Lines 159-160.

Point 9: at the beginning of the discussion narrate the key study findings.

Response 1: Thank you for your comment. Study Key findings are mentioned at the beginning of the discussion section “Being aged 40-49 years, having a moderate or a high perceived likelihood of being infected with COVID-19, and being in favour of mandatory vaccination were associated with COVID-19 vaccine acceptance.  However, females with a high educational level and those who had refused physician-recommended vaccine in the past were less likely to accept COVID-19 vaccination.”. Lines 220-224

Reviewer 2 Report

Understandig the factors associated with the acceptance of vaccination (particularly the COVID-19 Vaccine) is of utmost importance in the current pandemic context. Hence, author shud be commended for this work. 

Furthermore, it is important to understand the determinants of vaccine acceptance, particularly in women, from various geographical and cultural units such as Saudi Arabia.

I have no major comments; I have provided a few of my observations to consider below.

1. Please see if the below line can be paraphrased for better understanding:

"With this approval, another layer of protection that is not dependent on human behavior but rather on uptake is presented [4]."

2. Review the below line; seems to be a typo. Ebla outbreak happened in 2014-2016, See: https://www.cdc.gov/vhf/ebola/history/2014-2016-outbreak/index.html

"2104-15 West African Ebola outbreak indicates that"

3. Author mention this study is part of a larger study. I suggest author to cite the larger study or any papers published using the larger stusd, This will be helpful from the method perspective as well as for researchers. 

4. Relook at the below text in conclusion. Why P capital?

"This study suggests that Predictors of vaccine"

Author Response

Response to Reviewer 2 Comments

Point 1: Please see if the below line can be paraphrased for better understanding:

"With this approval, another layer of protection that is not dependent on human behavior but rather on uptake is presented [4]."

Response 1: Thank you for raising this comment. The sentence has been paraphrased to “. With the approval of the COVID-19 vaccine, a further layer of protection that is not dependent on human behavior (e.g., wearing a face mask) but rather on uptake becomes available” Lines 34-36.

Point 2: Review the below line; seems to be a typo. Ebla outbreak happened in 2014-2016, See: https://www.cdc.gov/vhf/ebola/history/2014-2016-outbreak/index.html

"2104-15 West African Ebola outbreak indicates that"

Response 1: Thank you for your comment. Indeed, the Ebola outbreak happened between 2014-2016. This was a typo and has been changed to” For instance, the 2014-16 West African Ebola outbreak indicates”. Lines 61-62

Point 3: Author mention this study is part of a larger study. I suggest author to cite the larger study or any papers published using the larger stusd, This will be helpful from the method perspective as well as for researchers. 

Response 1: Thank you for your comment. Related studies were cited

  1. Alfageeh, E.I.; Alshareef, N.; Angawi, K.; Alhazmi, F.; Chirwa, G.C. Acceptability of a COVID-19 Vaccine among the Saudi Population. Vaccines 2021, 9, 226, doi:10.3390/vaccines9030226.
  2. Qattan, A.M.N.; Alshareef, N.; Alsharqi, O.; Al Rahahleh, N.; Chirwa, G.C.; Al-Hanawi, M.K. Acceptability of a COVID-19 Vaccine Among Healthcare Workers in the Kingdom of Saudi Arabia. Front. Med. 2021, 8, 644300, doi:10.3389/fmed.2021.644300.
  3. Al-Hanawi, M.K.; Alshareef, N.; El-Sokkary, R.H. Willingness to Receive COVID-19 Vaccination among Older Adults in Saudi Arabia: A Community-Based Survey. Vaccines 2021, 9, 1257, doi:10.3390/vaccines9111257.

Point 4: Relook at the below text in conclusion. Why P capital?

"This study suggests that Predictors of vaccine"

Response 1: Thank you for your comment. this was a typo and has been changed to “This study suggests that predictors of vaccine”. Line 322
